# Multiview Fusion Using Transformer Model for Recommender Systems: Integrating the Utility Matrix and Textual Sources

**Thi-Linh Ho** [1,2,*], **Anh-Cuong Le** [1] **and Dinh-Hong Vu** [1]

1   Natural Language Processing and Knowledge Discovery Laboratory, Faculty of Information Technology, Ton Duc Thang University, Ho Chi Minh City 70000, Vietnam; leanhcuong@tdtu.edu.vn (A.-C.L.); vudinhhong@tdtu.edu.vn (D.-H.V.)
2   Faculty of Management Information System, Ho Chi Minh City University of Banking, Ho Chi Minh City 70000, Vietnam; linhht@hub.edu.vn (T.-L.H.)
*   Correspondence: hothilinh.st@tdtu.edu.vn

**Abstract:** Recommender systems are challenged with providing accurate recommendations that meet the diverse preferences of users. The main information sources for these systems are the utility matrix and textual sources, such as item descriptions, users' reviews, and users' profiles. Incorporating diverse sources of information is a reasonable approach to improving recommendation accuracy. However, most studies primarily use the utility matrix, and when they use textual sources they do not integrate them with the utility matrix. This is due to the risk of combined information causing noise and reducing the effectiveness of good sources. To overcome this challenge, in this study we propose a novel method that utilizes the Transformer Model, a deep learning model that efficiently integrates textual and utility matrix information. The study suggests feature extraction techniques suitable for each information source and an effective integration method in the Transformer model. The experimental results indicate that the proposed model significantly improves recommendation accuracy compared to the baseline model (MLP) for the Mean Absolute Error (MAE) metric, with a reduction range of 10.79% to 31.03% for the Amazon sub-datasets. Furthermore, when compared to SVD, which is known as one of the most efficient models for recommender systems, the proposed model shows a decrease in the MAE metric by a range of 34.82% to 56.17% for the Amazon sub-datasets. Our proposed model also outperforms the graph-based model with an increase of up to 108% in Precision, a decrease of up to 65.37% in MAE, and a decrease of up to 59.24% in RMSE. Additionally, experimental results on the Movielens and Amazon datasets also demonstrate that our proposed model, which combines information from the utility matrix and textual sources, yields better results compared to using only information from the utility matrix.

**Keywords:** recommender system; deep neural network recommender system; multiview; transformer model

## 1. Introduction

Recommender systems (RS) have become an essential tool for businesses to personalize their services by providing recommendations to their users. Accurately predicting user preferences has significant implications for businesses, as it allows them to increase customer satisfaction and loyalty. Despite the popularity of RS, it remains a challenging research problem, with researchers striving to enhance recommendation performance through innovative approaches.

RS are primarily developed using either collaborative filtering (CF) or content-based filtering techniques. The CF method utilizes ratings from similar users to predict ratings for an item, while the content-based approach predicts ratings for similar items based on the user's past ratings. Two well-known methods used in the CF approach are Neighborhood-based collaborative Filtering [1,2] and Matrix Factorization [3–7], which can be constructed

using only the information contained in the utility matrix. On the other hand, the content-based approach in such studies [2,8–11] is based on the simple observation that if a person enjoys item i, they are likely to enjoy similar products. The similarity between items can be calculated by referencing the utility matrix or item profiles found in other sources. However, content-based is limited in its ability to predict ratings for new items. To overcome this limitation, recent studies have combined both approaches using ensemble learning [2,12–17].

Traditionally, recommender systems rely on a single source of information, such as a user's historical purchase or viewing data, to generate recommendations. However, there is growing interest in using multiple sources of information to improve the accuracy and relevance of recommendations. A proposed strategy involves employing a multimodal approach, which pertains to combining data from different sources such as Refs. [18–22]. Additionally, there have been some recent studies [23–25] that employ neural networks to integrate information from multiple sources for ratings prediction. Hypergraphs provide a more flexible and powerful way to model complex interactions and dependencies between users and items in recommender systems, as demonstrated by previous research [26–28].

Despite the potential benefits of using a multimodal approach for recommender systems, there has been relatively little research to date on how to effectively combine all of these different sources of information into a unified model, especially on how to combine the utility matrix with other sources. We investigated the related studies and found that studies utilizing the utility matrix [29,30] tend to exclude other sources of information, possibly due to the effectiveness of the collaborative matrix factorization model for this particular information source. Therefore, when attempting to integrate other information sources, such as user reviews or item descriptions, it can cause interference and potentially reduce the accuracy of the system. Other multimodal methods that combine multiple sources often only utilize textual sources, while disregarding the utility matrix information, such as Refs. [25,31,32].

We believe that combining data from multiple sources, such as utility matrix, user profiles, and item descriptions, can lead to more personalized recommendations that reflect a user's unique interests and preferences. However, while the benefits of using a multimodal approach for recommender systems are clear, there are significant challenges that must be addressed. One major challenge is how to integrate the diverse sources of information effectively, without being affected by noise or conflicting signals that could lower the accuracy of the recommendations. This is particularly important because the information from different sources may complement each other or contain redundant information, and integrating them in a way that preserves their usefulness requires a careful balancing act.

Multimodal and multiview approaches provide numerous benefits that make them attractive for recommender systems. One significant advantage is their capability to capture multiple facets of user preferences and item characteristics, resulting in more precise and diverse recommendations. This is achievable because different information sources can provide complementary insights that may not be captured by a single modality or view [33–35].

Overall, our study makes several important contributions compared to other related studies, specifically as follows:

- Developing a novel model that combines views from both the utility matrix and textual sources, which utilizes feature extraction techniques from various information sources and a conversion algorithm to segment the feature vectors of each pair (user, item) into a sequence of token vectors, which serve as input for a classification model;
- Incorporating Transformer models into our approach for multimodal recommender systems, that serves as a strong tool for handling the challenge of integrating diverse sources of information in RSs, that can self-select features that avoid or minimize noise or conflicting signals, and it has the potential to significantly enhance the performance of RSs in practical settings;

- Conducting experiments on the MovieLens and Amazon datasets to verify the effectiveness of our proposed model.

The rest of the paper is organized as follows: Section 2 provides a summary of recent studies related to our work. Section 3 presents the conceptual and technical background for user-based and item-based recommender systems, the Transformer encoder architecture, matrix factorization, and feature extraction. In Section 4, we introduce our proposed model, which uses the transformer model on multiple views from diverse information sources. Section 5 presents the experimental results, comparisons, and discussions. Finally, in Section 6, we conclude our contribution.

## 2. Related Works

Recent research on collaborative filtering recommendation models primarily utilizes user rating scores and textual review data to compute and evaluate similarities between users or items. A model was introduced by Ghasemi, N. and Momtazi, S. [36] that identifies similar users by considering their reviews and ratings. Terzi, M. et al. [37] introduced a modification to the user kNN algorithm, measuring user similarity based on the similarity of their text reviews instead of ratings. Some studies have utilized rating scores to calculate user similarity and applied KNN algorithms to the results. For instance, Wang, Hua-Ming, Yu, and Ge used rating scores to compute users' similarity in Ref. [38], which was then input to KNN algorithms. Similarly, in Ref. [39], Cui and Bei-Bei calculated user similarity using rating scores, specifically cosine similarity and Pearson correlation similarity. The resulting similarity values were then used as input for the KNN algorithm. Kamali, P et al. utilized ratings to calculate Euclidean distance in Ref. [40]. In this study, we calculate the Cosine similarity between user–user or item–item using latent feature vectors generated from the utility matrix through matrix factorization.

Studies on collaborative filtering recommender systems that use user-based and item-based methods aim to enhance the accuracy of recommendations and address the sparsity challenge. Choudhury, S. S. [22] et al. introduced a cutting-edge trust matrix measure that incorporates user similarity and weighted trust propagation. Non-cold users underwent various models with a trust filter, while cold users derived an optimal score from personalized recommendations tailored to their preferences. Several research studies have proposed hybrid techniques that merge both user-based and item-based methods to enhance the performance of recommender systems. Zhang et al. [41] proposed a hybrid approach that combines user-based CF with item-based CF and content-based filtering to provide more accurate recommendations for new users. Liu et al. [42] suggest incorporating user embeddings as a novel method to improve the performance of SVD++-based collaborative filtering. The study's findings demonstrate that the proposed approach yields higher accuracy in rating prediction compared to traditional SVD++ and other advanced collaborative filtering techniques. Hasan, M., and Roy, F. [43] present an approach that incorporates trust and genre information into item–item collaborative filtering to address the cold-start problem. The results of the study demonstrate that the proposed method achieves higher accuracy in recommendations and better performance in addressing the cold-start problem compared to traditional collaborative filtering techniques. Duan, R. et al. [44] proposed a Review-Based Matrix Factorization method that combines review-based collaborative filtering and rating imputation to address the issue of sparsity in rating data for recommender systems. The method utilizes feature-level opinion mining of online review text to construct an item-topic rating matrix and populate the vacant values in the utility matrix, followed by matrix factorization to generate recommendations.

Recent studies have introduced neural network models to integrate information sources for recommender systems. He, X. et al. [23] proposed NCF, a collaborative filtering framework that employs neural networks and has the ability to generalize matrix factorization. To model non-linearities, they proposed a multi-layer perceptron that learns the user-item interaction function. Empirical evidence from extensive experiments supports the use of deeper layers of neural networks, which leads to better recommendation

performance. Nikzad–Khasmakhia, N. et al. [24] proposed BERTERS, a multimodal classification approach for expert recommender systems that combines text and graph modalities. BERT is used to convert text into vectors, while ExEm extracts features from the co-author network, which are concatenated with other features to generate a final representation of the candidate for the classifier. Yang, B. et al. [25] introduced an online video recommender system that utilizes multimodal fusion and relevance feedback. The system formulates video recommendation as finding the most relevant videos based on multimodal relevance, incorporating textual, visual, and aural features. It also incorporates relevance feedback to adjust weights within and among modalities based on the users' click-through data, and utilizes an attention fusion function to fuse multimodal relevance. Sun, F. et al. [31] introduced BERT4Rec, a sequential recommendation model that addresses limitations of left-to-right unidirectional models by employing bidirectional self-attention to model user behavior sequences. BERT4Rec uses the Cloze objective for training, predicting masked items in the sequence with joint conditioning on left and right context, leading to a bidirectional representation model that outperforms state-of-the-art sequential models in experiments. Qiu, G. et al. [32] proposed a text-aware recommendation model based on a multi-attention neural network model, which addresses the problem of finding reasons behind emotional expressions in texts. The model uses modified LDA and paragraph vector learning for text vector representation, captures context information through Bi-LSTM layer, and employs CNN layer for emotional factor classification and prediction.

Recent studies have mentioned hypergraphs as a flexible and powerful method for modeling complex user–item interactions and dependencies in multimodal recommender systems. K. Pliakos and C. Kotropoulos [26] developed a novel approach for simultaneous image tagging and geo-location prediction using hypergraph learning. The method is enhanced by incorporating group sparsity constraints and utilizing diverse forms of information, including social data, image metadata, and visual similarities. Xia, X. et al. [27] introduced DHCN, a dual channel hypergraph convolutional network, to enhance session-based recommendation. They incorporated self-supervised learning into the network's training to improve hypergraph modeling and maximize mutual information between the session representations learned through both channels in DHCN. GraphRec, a new graph neural network framework, was introduced by Fan, W. et al. [28] for social recommendations, incorporating an approach that integrates interactions and opinions in the user-item graph and coherently models two graphs with varying strengths.

Although using multiple sources of information in recommender systems has potential benefits, research on effectively combining them is limited, particularly when it comes to integrating the utility matrix with other sources. Our investigation revealed that studies utilizing the utility matrix tend to exclude other sources of information, potentially leading to interference and reduced accuracy when integrating other sources such as user reviews or item descriptions. We propose a multiview transformer recommendation model that integrates data from various sources, including the utility matrix, to provide personalized recommendations that reflect a user's preferences. However, effective integration of diverse sources presents significant challenges, such as avoiding noise or conflicting signals. Our study demonstrates the potential of using a multimodal approach for RS and proposes a transformer model to enhance the performance of recommender systems.

## 3. Background

### 3.1. User-Based Recommender Systems

The fundamental concept is to identify users who share similar preferences with the target user $u$, and utilize this information to predict ratings for the unrated items by $u$.

The cosine distance is a measure of similarity between two vectors, often used in recommender systems to calculate the similarity between items or between users. In the context of recommender systems, the cosine distance is often used with the cosine similarity measure to compute the similarity between items or between users based on their rating patterns.

The cosine similarity between two vectors $u$ and $v$ is calculated as the cosine of the angle between them, which can be expressed as the dot product of the two vectors divided by the product of their magnitudes. The cosine distance is simply the complement of the cosine similarity, i.e., 1 minus the cosine similarity. The cosine distance between two vectors u and v is therefore calculated as 1 minus the cosine of the angle between them. The cosine distance ranges from 0 (indicating perfect similarity) to 2 (indicating maximum dissimilarity). In recommender systems, the cosine distance is often used as a dissimilarity measure, so items or users with a smaller cosine distance are considered more similar.

The cosine similarity between two vectors $u$ and $v$ is defined as the cosine of the angle between the two vectors. It is a measure of similarity between two non-zero vectors in an inner product space and can be calculated using Equation (1).

$$sim(u,v) = \frac{u.v}{||u|| ||v||} \tag{1}$$

where:

- $u.v$ is the dot product of vectors $u$ and $v$;
- $||u||$ and $||v||$ are the Euclidean norms of vectors $u$ and $v$, respectively.

By computing the dot product of two vectors and dividing it by the product of their magnitudes, we can determine the cosine similarity between them. This similarity score falls between $-1$ and 1, where 1 signifies complete similarity, 0 indicates orthogonality, and $-1$ represents complete dissimilarity. Equation (2) is utilized to predict the rating that a user $u$ is likely to assign to an item $j$ [45].

$$\check{R}_{u,j} = \bar{R}_u + \frac{\sum_{v \in KNN}(R_{v,j} - \bar{R}_v).sim(u,v)}{\sum_{v \in KNN}|sim(u,v)|} \tag{2}$$

where $KNN$ is a set of k nearest neighbors of user $u$, $R_{v,j}$ is the rating of user $v$ on item $j$, $\bar{R}_u$ and $\bar{R}_v$ are the average ratings of users $u$ and $v$ respectively, and $sim(u,v)$ denotes the similarity between users $u$ and $v$.

### 3.2. Item-Based Recommender Systems

The first step in predicting the ratings that user $u$ would give to target item $j$ is to find a set $S$ of items that have the most similarity to $j$. This requires calculating similarity functions between the columns of a matrix containing user-item ratings, to locate items with comparable attributes. Additionally, inverse item frequency, which refers to the frequency of ratings that users give to those items, is taken into account when producing similarities between items. In the cosine similarity method, Equation (3) can be used to compute the similarity between items $i$ and $j$ with inverse item frequency, according to Ref. [46].

$$sim(i,j) = \frac{\sum_{u \in (U_i \cap U_j)}(R_{u,i} \times \log(n/f_u)) \times (R_{u,j} \times \log(n/f_u))}{\sqrt{\sum_{u \in U_i}(R_{u,i} \times \log(n/f_u))^2}\sqrt{\sum_{u \in U_j}(R_{u,j} \times \log(n/f_u))^2}} \tag{3}$$

The users who have rated items $i$ and $j$ are referred to by the sets $U_i$ and $U_j$ respectively. $R_{u,i}$ and $R_{u,j}$ denote the ratings given to item $i$ and $j$, respectively, by user $u$. The inverse item frequency of a user is represented by the function $\log(n/f_u)$, where $n$ represents the total number of items in the system and $f_u$ indicates the number of items rated by user $u$. If user $u$ has rated all items, the inverse item frequency value is 0. Equation (4) can be employed formally to estimate the rating that the target user $u$ would assign to the target item $j$. The equation utilizes the set of k' most similar items to item $j$, represented by $MSI$.

$$\check{R}_{u,j} = \frac{\sum_{i \in MSI} sim(i,j) \times R_{u,i}}{\sum_{i \in MSI}|sim(i,j)|} \tag{4}$$

### 3.3. The Transformer Encoder Architecture

The Transformer encoder architecture (illustrated in Figure 1), introduced in Ref. [47], is widely used in natural language processing (NLP) for sequence-to-sequence tasks such as machine translation and language modeling. It is known for its ability to effectively capture long-range dependencies and its parallelizability.

The key components of the Transformer encoder architecture include input embeddings, positional encoding, multi-head self-attention, layer normalization, position-wise feed-forward networks, residual connections, and layer stacking.

Input embeddings represent the meaning of input tokens and are learned during training. Positional encoding is added to address the lack of inherent positional information in the Transformer architecture, allowing the model to capture the sequential order of tokens in the input sequence.

The multi-head self-attention mechanism is at the heart of the Transformer encoder, enabling the model to weigh the importance of different tokens and capture long-range dependencies efficiently. Layer normalization is applied after self-attention to normalize the outputs, aiding in training stability. Position-wise feed-forward networks capture complex interactions between tokens, and residual connections help with gradient flow and alleviate the vanishing gradient problem.

The Transformer encoder architecture typically consists of multiple identical layers stacked on top of each other, with the outputs of each layer fed as inputs to the subsequent layer, allowing the model to capture hierarchical representations of the input sequence.

In summary, the Transformer encoder architecture is characterized by its self-attention mechanism, layer normalization, and residual connections, which work together to make it a powerful architecture for sequence-to-sequence tasks in NLP.

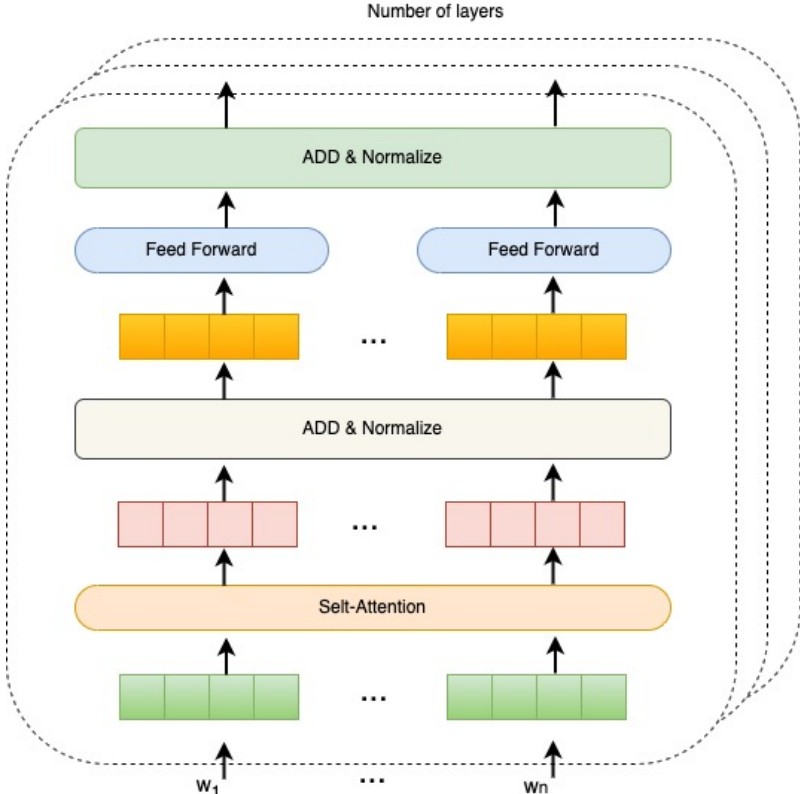

**Figure 1.** The Transformer encoder architecture.

### 3.4. Matrix Factorization

The Matrix Factorization method, as stated in Ref. [48], utilizes the utility matrix to identify latent features that represent users and items. If we are given a rating matrix

of shape $(n \times m)$, which presents the ratings of n users on m items, the aim of Matrix Factorization is to decompose this matrix into two thin matrices, $P$ and $Q$, both with a shape of $n \times f$ and $m \times f$, respectively (shown in Equation (5)). The value of $f$ indicates the number of important latent factors contained in the matrices. Figure 2 depicts the decomposition process of matrix $R$ into matrices $P$ and $Q$.

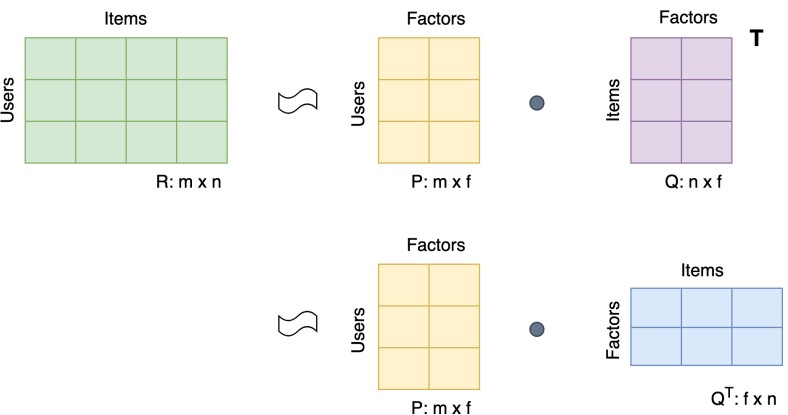

**Figure 2.** The process of breaking down the utility matrix into matrices of latent factors.

$$R \approx P \cdot Q^T \qquad (5)$$

The representation of the formula used to predict the rating $r$ for a pair consisting of user $u$ and item $i$ is expressed as Equation (6), where the preference of users for items is denoted by the product of the user vector $p_u$ and the transpose of the item vector $q_i$, both of which have $f$ dimensions.

$$\hat{r}_{u,i} = p_u \cdot q_i^T \qquad (6)$$

### 3.5. Feature Extraction

### 3.5.1. Term Frequency-Inverse Document Frequency (TF-IDF)

TF-IDF tackles a specific issue that may not occur frequently in our corpus but is immensely significant. The TF-IDF value escalates in proportion to the frequency of a word's occurrence in a document, but it reduces in relation to the number of documents in the corpus containing the word [49]. TF-IDF consists of two sub-parts, which are:

- Term Frequency (TF), which is calculated by Equation (7), represents the frequency of occurrence of a term within the entire document, which can be interpreted as the likelihood of discovering a particular word in the document. It is computed as the number of occurrences of a word $w_i$ in a review $r_j$, relative to the total number of words in the review $r_j$;
- Inverse Document Frequency (IDF) is a metric that evaluates the frequency of a term across the documents in a corpus, as described by Equations (8) and (9). It emphasizes rare words that appear in only a few documents throughout the corpus and results in a high IDF score. The value of IDF is obtained by log-normalizing the ratio of the total number of documents $D$ in the corpus to the number of documents that contain the term $t$ .

$$\textit{tf-idf(t,d)} = tf(t,d) \times idf(t) \qquad (7)$$

If *smooth_idf = False*, the formula for calculating $idf(t)$ is the following:

$$idf(t) = \log \frac{n}{df(t)} + 1 \qquad (8)$$

If *smooth_idf = True* (this prevents zero division), the formula for calculating *idf(t)* is the following:

$$idf(t) = \log \frac{n+1}{df(t)+1} + 1 \tag{9}$$

where *n* is the total number of documents in the corpus and $df(t)$ is the document frequency of *t*; the document frequency is the number of documents in the corpus that contain the term *t*.

### 3.5.2. Bidirectional Encoder Representations from Transformers (BERT)

BERT [50] is a language model that is pre-trained and is extensively used for various natural language processing tasks. Due to its foundation on the Transformer architecture, which utilizes a multi-head attention mechanism, BERT is highly proficient in representing and encoding textual information. Initially, text is represented by a sequence of tokens and is initialized as one-hot vectors. Subsequently, processing the layers of the Transformer involves computing the attention of words with words adjacent to them to generate the word representation in the subsequent layers. In the BERT model, additional positional information is utilized. BERT is then trained by masking certain words (using MASK) during the input and predicting them in the output layer.

Incorporating BERT as a preprocessing tool has enabled numerous NLP tasks to achieve state-of-the-art performance. Therefore, in our research, we also leverage BERT to represent textual information such as reviews and descriptions. Specifically, we employ two variations of BERT: BERTBASE (L = 12, H = 768, A = 12, total parameters = 110 M) and BERTLARGE (L = 24, H = 1024, A = 16, total parameters = 340 M). For generating word vectors in our study, we opt to use BERTBASE.

## 4. The Proposed Model

### 4.1. Our Multiview Transformer Model for Recommendation

Multiview fusion is the integration of information from multiple sources or views, such as images, text, audio, video, and other types of data, to create a comprehensive and accurate representation of a given situation or event. Multiview fusion techniques are widely utilized in the fields of machine learning and artificial intelligence to enhance the performance of models that deal with complex and diverse data. By combining information from different views, these models can gain a deeper understanding of underlying patterns and relationships in the data, leading to improved predictions and decision-making.

Early fusion and late fusion are two approaches for multiview fusion. Early fusion combines features or representations of different views at the input level, transforming them into a common feature space before feeding them into the fusion model. It is suitable for highly correlated views that provide complementary information. In contrast, late fusion involves fusing outputs of individual models trained on each modality separately. Each view is processed using a specific model, and the outputs are then combined using a fusion mechanism. Late fusion is used for views that are not highly correlated and provide unique, independent information.

Our Multiview Transformer recommendation model, described in Algorithm 1, utilizes an early fusion approach. The architecture of our proposed model for the recommendation problem is depicted in an overview in Figure 3. It comprises three main components: the first includes modules for extracting and representing features from various information sources of users and items, the second involves converting the concatenated vector into n segments (described in Algorithm 2), and the third consists of a transformer module for the prediction task.

---

**Algorithm 1** Multiview Transformer Model For Recommendation

---

1: **Training Stage:**
2: Input: The data set contains multiview of a set of users and items.
3: Output: A model for the rating
4: Feature Extraction: involves performing feature extraction for each modality, resulting in a vector of features for each modality. For a given user, a set of k feature vectors denoted by $V_u = v_{u1}, \cdots, v_{uk}$ is obtained, and for a given item, a set of $j$ feature vectors denoted by $V_i = v_{i1}, \cdots, v_{ij}$ is obtained.
5: Feature Conversion: involves combining feature vectors from Step 1 and segment it into a sequence of token vectors as input to the classification model.
6: Classification: building the Transformer Model for the Prediction task (named as TMP)
   - Feeding n tokens from Step 2 into a transformer encoder model is used for the task of rating generation (i.e., prediction),
   - Training prediction model.
7: **Inference stage:**
8: A given user-item pair (u, i).
9: Performing the feature extraction task to obtain feature vectors for user u and item i. These feature vectors include $V_u = v_{u1}, \cdots, v_{uk}$ and $V_i = v_{i1}, \cdots, v_{ij}$.
10: Combining $V_u$ and $V_i$, then using the Feature Conversion algorithm to generate n segments.
11: Feeding n tokens at Step 2 as input features for the TMP for generating the output (i.e., the rating).

---

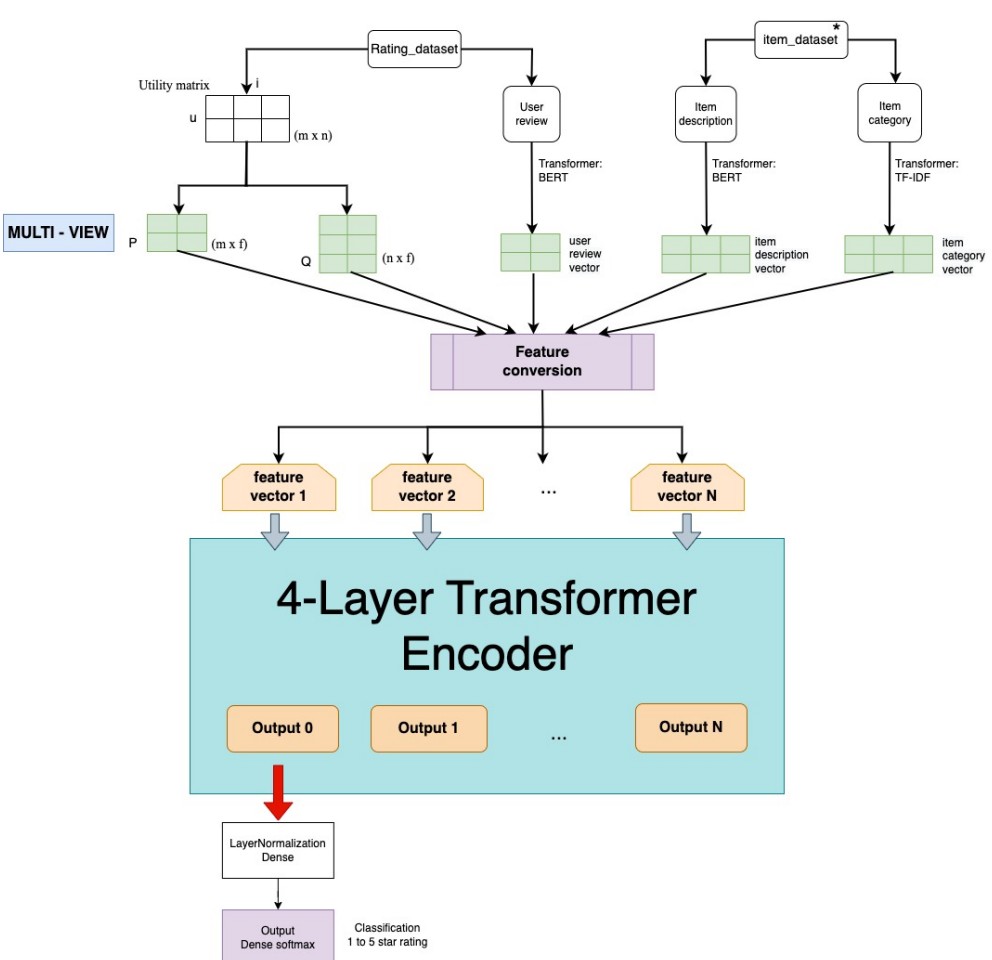

**Figure 3.** Multiview fusion with transformer model for recommender systems. (The symbol * indicates the amount of item attribute information used in the model may vary depending on the dataset and the intended use of the organization.)

*4.2. The Feature Extraction Algorithm*

In our proposed model, the component responsible for extracting and representing features is crucial. We generate a variety of features to represent users and items by utilizing information from four distinct sources: the utility matrix, user reviews, item descriptions, and item categories. The details of our approach are outlined below:

We use the matrix factorization technique to extract latent features of users and items from the given utility matrix, which we refer to as Rating-Feature. This technique involves decomposing the utility matrix of size $(m \times n)$ containing ratings for $m$ users and $n$ items into two lower rank matrices. Here, $f$ refers to the number of latent factors or features that represent the users and items in the decomposed matrices. To perform factorization, we select the number of latent factors and use the Alternating Least Square algorithm. The user and item matrices are initialized with random values and iteratively optimized to minimize the difference between predicted and actual ratings. This process alternates between fixing the user matrix and updating the item matrix until convergence, resulting in two lower-dimension matrices of $m \times f$ and $n \times f$ dimensions for the user and item matrices, respectively.

1.  The user features are represented by a matrix of size $(m \times f)$, where each row corresponds to a specific user and contains a vector of $f$ features. This matrix is considered as a view of the user, meaning that it represents the user's characteristics;
2.  The item features are represented by a matrix of size $(n \times f)$, where each item is represented by a vector of $f$ features. Each row in this matrix corresponds to a specific item and is considered a view of that item.

For the Review-Feature, we employ a pre-trained BERT model to generate a vector representation of text reviews written by users for items in the (user $u$, item $i$) pair. This vector representation is considered a view of user $u$.

For the Description-Feature, we utilize a pre-trained BERT model to generate a vector representation of the text description provided for item $i$. This vector representation is considered a view for item $i$.

The Category-Feature involves using the list of categories to which item $i$ belongs to generate a vector representation. TF-IDF measures the significance of words in a document (which in our study is a list of item categories) by considering their frequency in the document and the frequency of the word in the corpus. Since BERT is better suited for extracting features from complete sentences, we use TF-IDF instead of BERT to convert these category features into a vector representation. This vector representation is also considered a view for item $i$.

Some of the feature types that will be extracted are denoted as the following:

- Rating-Feature, which pertains to the user $u$ and item $i$, and is represented as Rat($u$) and Rat($i$) respectively;
- Review-Feature, which pertains to the user $u$ and is represented as Rev($u$);
- Description-Feature, which pertains to the item $i$ and is represented as Des($i$);
- Category-Feature, which pertains to the item $i$ and is represented as Cat($i$).

*4.3. The Feature Conversion Algorithm*

Note that after obtaining the feature presentation as a sequence of token vectors in the steps above, we design a Transformer model for the ratings classification.

---

**Algorithm 2** The Feature Conversion Algorithm

---

1: For each pair (user u, item i), we use the Feature Extraction module to generate various types of features represented as corresponding feature vectors:

- Feature vectors for user u: $fu_1, fu_2$
- Feature vectors for item i: $fi_1, fi_2, fi_3$

2: Transform features:
   - We firstly combine these feature vectors and obtain the united vector:

$$F1 = Concat(fu_1, fu_2, fi_1, fi_2, fi_3)$$

   - This obtained vector $F1$ is then fed into feedforward layers (denoted by FFLayers):

$$F2 = FFLayers(F1)$$

   - The vector $F2$ is then segmented to create a sequence of token vectors that will be fed into the Transformer model for the task of ratings prediction.

$$(token_1, ..., token_k) = segment(F2)$$

---

## 5. Experimental Results

### 5.1. The Dataset Summary

In order to assess the efficacy of our proposed model, we employed the MovieLens dataset and various Amazon sub-datasets. To enhance the quality of our analysis, we preprocessed these datasets to exclude users who infrequently rated items and items that rarely received the ratings from the community.

We utilized the Amazon dataset for our experiments, which can be accessed at the website https://jmcauley.ucsd.edu/data/amazon/ (accessed on 30 October 2022). Specifically, we employed sub-datasets including Electronic, Video Games, and Toys and Games. To enhance the scope of our analysis, we incorporated various types of information, such as rating scores, user reviews, item categories, and item descriptions.

For our experiments, we utilized the MovieLens dataset, which can be accessed through the website https://grouplens.org/datasets/movielens (accessed on 30 October 2022). The dataset provided us with various information such as user and movie identification, rating scores, and movie genres, which we leveraged in our analysis.

Table 1 contains a description of the MovieLens and Amazon datasets:

- The MovieLens dataset includes 77,763 ratings of 27,041 users and 8203 movies;
- The Electronic dataset includes 80,472 ratings of 1042 users and 21,200 items;
- The Video Games dataset includes 98,769 ratings of 2291 users and 24,708 items;
- The Toys and Games dataset includes 68,102 ratings of 5462 users and 3028 items.

**Table 1.** A summary of the datasets.

| Dataset | Ratings | Users | Items |
|---|---|---|---|
| MovieLens | 77,763 | 27,041 | 8203 |
| Amazon: Toys and Games subcategory | 68,102 | 5462 | 3028 |
| Amazon: Video and Games subcategory | 98,769 | 2291 | 24,708 |
| Amazon: Electronic subcategory | 80,472 | 1042 | 21,200 |

### 5.2. Evaluation Metrics

5.2.1. Mean Absolute Error (MAE)

The MAE is a popular metric because it ensures consistency in units between error and predicted target values, and treats all errors equally. The MAE score can be calculated

using Equation (10), which involves summing up the absolute errors and dividing the result by the total number of observations [51].

$$MAE = \frac{1}{n} \sum_{i=1}^{n} |y_1 - \hat{y}_i|^2 \tag{10}$$

### 5.2.2. Root Mean Square Error (RMSE)

The RMSE is a measure that quantifies the discrepancy between the actual and predicted ratings. It is calculated by taking the mean value of the squared differences between the actual and predicted ratings, and finding the square root of that result, making it useful when significantly large errors are undesirable [52]. Equation (11) displays the RMSE calculation formula, in which $d_i$ and $\hat{d}_i$, respectively, indicate the actual and predicted ratings, while $n$ denotes the total number of ratings.

$$RMSE = \sqrt{\frac{1}{n} \sum_{i=1}^{n} (d_i - \hat{d}_1)^2} \tag{11}$$

### 5.2.3. Precision

To evaluate the numerical precision of the predicted ratings, we employ a metric called Precision, which is computed using the Equation (12). In this equation, $TP$ refers to the number of predicted ratings that agree with the corresponding test ratings, while $FP$ corresponds to the number of predicted ratings that differ from the test ratings [53].

$$Precision = \frac{TP}{TP + FP} \tag{12}$$

### 5.3. Experimental Setups for Data

To conduct the experiments, we divided the dataset into smaller sets for training, validation, and testing, and the details are presented in Table 2. Specifically, the number of ratings for the training set, validation set, and testing set of each dataset are as follows:

- For the MovieLens dataset, there are 62,211 ratings for training, 3110 ratings for validation, and 12,442 ratings for testing;
- For the Toys and Games dataset, there are 54,482 ratings for training, 2724 ratings for validation, and 10,896 ratings for testing;
- For the Electronic dataset, there are 64,378 ratings for training, 3218 ratings for validation, and 12,876 ratings for testing;
- For the Video and Games dataset, there are 79,016 ratings for training, 3950 ratings for validation, and 15,803 ratings for testing.

**Table 2.** Setting up the data for the experiments.

| The Dataset | Trained Ratings | Validated Ratings | Tested Ratings |
|---|---|---|---|
| MovieLens | 62,211 | 3110 | 12,442 |
| Amazon-Toys and Games | 54,482 | 2724 | 10,896 |
| Amazon-Electronic | 64,378 | 3218 | 12,876 |
| Amazon-Video and Games | 79,016 | 3950 | 15,803 |

### 5.4. Experimental Setups for Models

The goal of our research is to demonstrate that combining different views using the Transformer model is more efficient than using only one view, such as the Utility Matrix, and is better than using the SVD model for the Utility Matrix. It is important to empirically prove this hypothesis using the same established configuration. Therefore, we choose the test configuration without paying full attention to choosing the configuration that gives

the best results. The parameter values were chosen based on previous studies and were experimented with several different values to choose the best configuration within that set. For the magnitude of the attribute vectors, we have set the following:

- Because we use BERT for representing user reviews in our dataset, the magnitude of the resulting attribute vector is 768 dimensions, which is equal to the magnitude of the output vector of the BERT-base model;
- To represent the latent attributes of users and items using the factorization matrix method, we tested different magnitudes of the latent feature vector, including 20, 50, and 100. After evaluating the performance on the validation set, we selected the magnitude of 50 as the best choice;
- When using the TF-IDF measure for attribute representation, we utilized the set of all tokens in our data. Therefore, the corresponding magnitude of these representation vectors will be equal to the number of keywords.

Given that $fu_1$, $fi_1$, $fi_2$, $fu_2$, and $fi_3$ represent user latent features, item latent features, item genre/category features, user review features, and item description features, respectively. Our proposed Multiview Transformer recommendation model is set up with dimensional parameters for each input feature vector, as shown in Table 3.

- For the Movielens dataset, the dimensions of $fu_1$, $fi_1$, and $fi_2$ are 50, 50, and 20, respectively;
- For the Amazon-Toys and Games dataset, the dimensions of $fu_1$, $fi_1$, $fi_2$, $fu_2$, and $fi_3$ are 50, 50, 321, 768, and 768, respectively;
- For the Amazon-Electronics dataset, the dimensions of $fu_1$, $fi_1$, $fi_2$, $fu_2$, and $fi_3$ are 50, 50, 944, 768, and 768, respectively;
- For the Amazon-Video and Games dataset, the dimensions of $fu_1$, $fi_1$, $fi_2$, $fu_2$, and $fi_3$ are 50, 50, 16,978, 768, and 768, respectively.

**Table 3.** Setting up the parameters for the multiview transformer RS.

| The Dataset | The Dimensions of User Latent Features | The Dimensions of Item Latent Features | The Dimensions of Item Genre/Category Features | The Dimensions of User Review Features | The Dimensions of Item Description Features |
|---|---|---|---|---|---|
| MovieLens | 50 | 50 | 20 | | |
| Amazon-Toys and Games | 50 | 50 | 321 | 768 | 768 |
| Amazon-Electronic | 50 | 50 | 944 | 768 | 768 |
| Amazon-Video and Games | 50 | 50 | 16,978 | 768 | 768 |

For the Multiview selection, we designed different experiments to investigate the effect of using different views of the data, namely:

- Experiment with data represented by 2 views, representing users and items, generated from the utility matrix;
- Experiment with data represented by all views. For MovieLens data, there are 3 views in all, of which 2 are from the utility matrix and 1 view is Genre; For Amazon data, there are 5 views in all, of which 2 are from the utility matrix and 3 are from user review, item description, and item category.

For the ratings prediction model, we have designed the following models:

- We have conducted experiments with different configurations of the Transformer model in our proposed approach, including varying numbers of layers (2, 4, or 6) and hidden state values (20 and 50). After evaluating their performance on the validation set, we selected the optimal configuration of (Layers = 4, Hidden state = 20, Heads = 4);
- We have also designed two baseline models, namely the MLP (Feedforward Neural Network) and the Gated Recurrent Units (GRU), to demonstrate that the Transformer model performs better;

- In addition, we have tested a strong baseline model, the Singular Value Decomposition (SVD) on the utility matrix, which is known to be one of the most effective models for the recommender system (RS) problem.

We trained our model for 50 epochs and conducted the experiment 3 times for each model. As there was only a negligible difference between the experimental runs, we chose to include the best result from the experiment in Table 4.

Within the graph-based recommendation model, we have constructed a graph that encompasses two distinct node types: user and item.

- For the Amazon dataset, we incorporated category and description as attributes for the item nodes, while the user nodes were enriched with the review attribute;
- For the MovieLens dataset, the item nodes were augmented with the genre attribute;
- We established connections between the user and item nodes in the graph to model the interactions between them. To represent the strength of these interactions, we assigned ratings as the weight property of the edges. This allowed us to reflect the degree of preference that each user showed for each item, which is an important factor in our recommendation system.

**Table 4.** The experimental results on the MovieLens and Amazon datasets

| Models | MAE | RMSE | Precision |
|---|---|---|---|
| MovieLens: | | | |
| MLP RS [three views] | 1.261 | 1.677 | 51.03% |
| GRU RS [three views] | **0.852** | **1.062** | 65.39% |
| Transformer-based RS [two views] | 0.973 | 1.230 | 67.11% |
| Transformer-based RS [three views] | 0.964 | 1.216 | **67.16%** |
| SVD RS | 1.562 | 2.293 | 47.75% |
| Amazon: Toys and Games: | | | |
| MLP RS [five views] | 0.527 | 0.776 | 86.65% |
| GRU RS [five views] | 0.729 | 0.889 | 86.31% |
| Transformer-based RS [two views] | 0.570 | 1.09 | 86.85% |
| Transformer-based RS [five views] | **0.445** | **0.743** | **92.07%** |
| SVD RS | 0.844 | 1.120 | 68.06% |
| Amazon: Video and Games: | | | |
| MLP RS [five views] | 0.729 | **0.999** | 73.74% |
| GRU RS [five views] | 0.941 | 1.174 | 76.89% |
| Transformer-based RS [two views] | 0.842 | 2.062 | 77.09% |
| Transformer-based RS [five views] | **0.689** | 1.471 | **83.31%** |
| SVD RS | 1.572 | 1.254 | 63.99% |
| Amazon: Electronic: | | | |
| MLP RS [five views] | 0.628 | **0.914** | 83.22% |
| GRU RS [five views] | 0.78 | 1.007 | 85.80% |
| Transformer-based RS [two views] | 0.606 | 1.362 | 85.94% |
| Transformer-based RS [five views] | **0.554** | 1.195 | **87.83%** |
| SVD RS | 0.85 | 1.159 | 69.59% |

*5.5. Results*

We evaluated the performance of our proposed Multiview transformer recommendation model by comparing it with the SVD, MLP, and GRU recommendation model on the MovieLens and Amazon datasets. Our experimental findings validate two hypotheses:

- Combining information from the utility matrix and other textual features using the Transformer model results in better outcomes compared to relying solely on features generated from the utility matrix;
- The Transformer model outperforms other models (specifically, we compared it with GRU and MLP) in terms of yielding better results.

Table 4 presents the experimental findings for four datasets, which include The MovieLens dataset and three Amazon sub-datasets (Toys and Games, Video and Games, and Electronic). Regarding the first hypothesis, our proposed model outperforms SVD (SVD RS), using only the information derived from the utility matrix, in all three metrics (MAE, RMSE, and Precision) on all four datasets. Regarding the second hypothesis, the Transformer model achieves the best Precision results, compared to the GRU (GRU RS) and MLP (MLP RS) models. The Transformer model also outperforms the other models in the MAE and RMSE metrics for most datasets, except for the MovieLens dataset, where the GRU model produces the best results in both the MAE and RMSE metrics. Similarly, for the Amazon-Electronic dataset, the MLP model yields the best RMSE.

Specifically, during the experiments conducted on the Movielens dataset, our proposed transformer recommendation model integrated three views, which were user latent features, item latent features, and item genre features. The experimental results obtained were MAE, RMSE, and Precision metrics of 0.964, 1.216, and 67.16%, respectively. Compared to the MLP, GRU, and SVD recommendation models, the proposed transformer recommendation model exhibited the best Precision (Shown in Figure 4). Furthermore, our transformer recommendation model provided slightly better results when integrating three views than when integrating only two views.

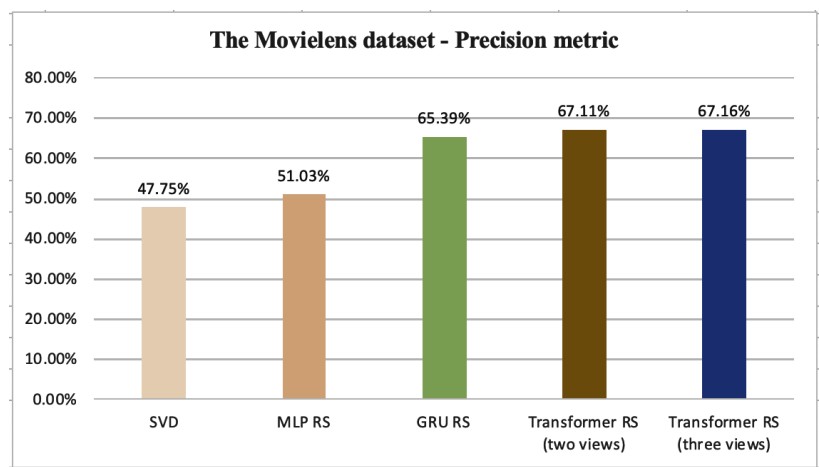

**Figure 4.** The experimental results for the Precision metric with the Movielens dataset.

When conducting experiments on the Amazon-Toys and Games dataset, our proposed transformer recommendation model integrated five views, which were the user latent features, item latent features, item category features, item description features, and features extracted from user reviews. The model provided better results in terms of MAE, RMSE, and Precision metrics of 0.445, 0.743, and 92.07%, respectively, compared to recommendation models based on MLP, GRU, and SVD (Shown in Figure 5). Furthermore, our proposed transformer recommendation model provided better results when integrating five views than when integrating only two views containing user latent features and item latent features.

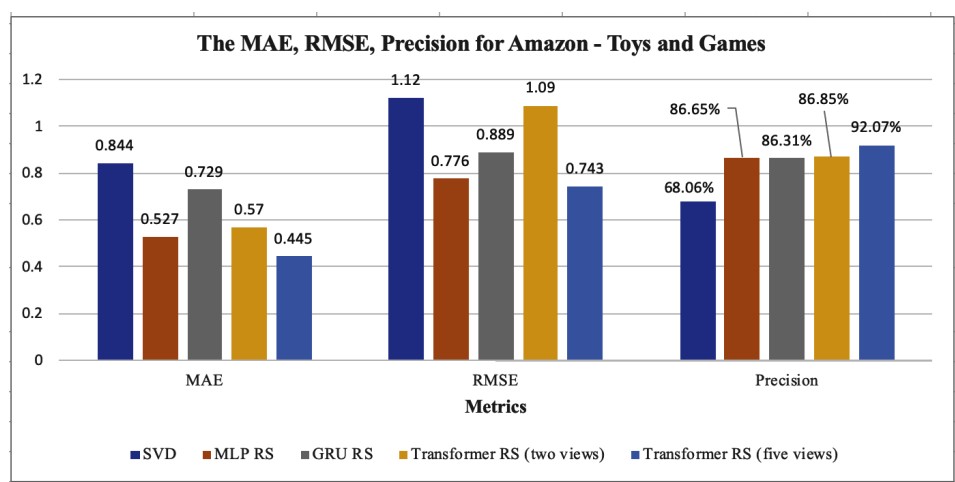

**Figure 5.** The experimental results for the MAE, RMSE, and Precision with the Amazon-Toys and Games dataset.

When conducting experiments on the Amazon-Video and Games dataset, our proposed transformer recommendation model, which integrated five views, also gave the MAE, RMSE, and Precision metrics of 0.689, 1.471, and 83.31%, respectively. Compared to recommendation models based on MLP, GRU, and SVD, our proposed transformer recommendation model exhibited the best results in terms of MAE and Precision (shown in Figure 6). Moreover, our proposed transformer recommendation model provided better results when integrating five views than when integrating only two views containing user latent features and item latent features.

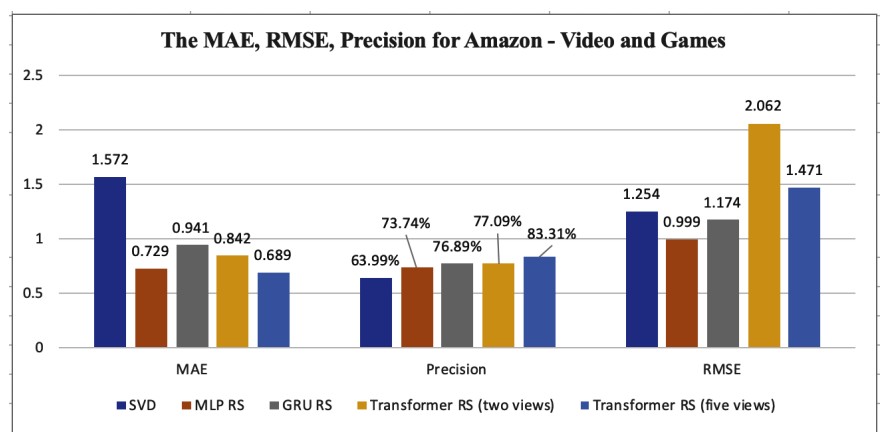

**Figure 6.** The experimental results for the MAE, Precision with the Amazon-Video and Games dataset.

We also conducted experiments on the Amazon-Electronics dataset, which resulted in MAE, RMSE, and Precision metrics of 0.554, 1.195, and 87.83%, respectively, demonstrating the effectiveness of our proposed transformer recommendation model that integrates five views. Compared to recommended models based on MLP, GRU, and SVD, our proposed transformer recommendation model exhibited the best results in terms of MAE and Precision (shown in Figure 7). Additionally, the proposed transformer recommendation model performed better when integrating five views rather than only two views containing user latent features and item latent features.

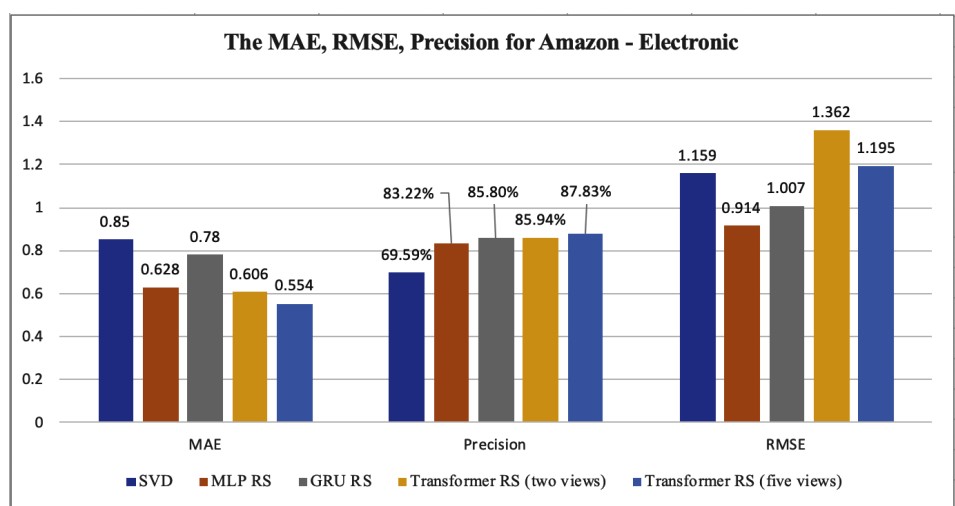

**Figure 7.** The experimental results for the MAE, Precision with the Amazon-Electronic dataset.

In addition to comparing our proposed model with SVD-based and DNN-based (MLP and GRU) recommendation models, we have conducted experiments using the graph-based recommendation model on the same train and test sets derived from the MovieLens and Amazon sub-datasets. Our experiments provide evidence that our proposed recommendation model surpasses the performance of the graph-based recommendation model. Specifically, the results from Table 5 illustrate that our proposed model achieves higher recommendation accuracy, with a Precision metric increase ranging from 11.14% to 108%, a MAE metric decrease ranging from 31.19% to 65.37%, and a RMSE metric decrease ranging from 15.56% to 59.24%.

**Table 5.** The comparison between our proposed model and a graph-based RS.

| Dataset | Methods | Number of Views | MAE | RMSE | Precision |
|---|---|---|---|---|---|
| MovieLens | Graph | Three views | 1.401 | 1.908 | 60.43% |
| | Our proposed model: Transformer | Three views | 0.964 | 1.216 | 67.16% |
| Amazon: Toys and Games | Graph | Five views | 1.285 | 1.823 | 60.30% |
| | Our proposed model: Transformer | Five views | 0.445 | 0.743 | 92.07% |
| Amazon: Video and Games | Graph | Five views | 1.305 | 1.742 | 56.01% |
| | Our proposed model: Transformer | Five views | 0.689 | 1.471 | 83.31% |
| Amazon: Electronic | Graph | Five views | 1.542 | 1.863 | 42.25% |
| | Our proposed model: Transformer | Five views | 0.554 | 1.195 | 87.83% |

## 6. Conclusions

In this study, we propose a new multimodal approach (i.e., multiview fusion) for the recommender systems problem. Our method integrates various information sources, including the utility matrix and textual sources, which previous studies have not been able to solve. Our model is based on the Transformer Model and feature extraction methods for each information source.

Our experimental results show that the proposed model performs better than both the SVD method and the baseline model (MLP), which only use user and item representation based on the utility matrix. In summary, the experimental results show that our proposed model achieved higher recommendation accuracy compared to the baseline model (MLP)

in terms of Precision, with increases ranging from 6.03% to 35.48% for the Amazon and MovieLens datasets. The proposed model also demonstrated improved accuracy in terms of MAE, with reductions ranging from 10.79% to 31.03% for the Amazon datasets. Furthermore, compared to SVD, known as one of the most effective models for RSs, the proposed model showed an increase in Precision ranging from 26.21% to 40.65%, and in MAE ranging from 34.82% to 56.17% for the Amazon datasets.

Moreover, the experimental results obtained from the MovieLens and Amazon datasets provide evidence that our proposed model, which integrates information from both the utility matrix and textual sources, outperforms using only the information from the utility matrix in terms of recommendation accuracy. Moreover, we emphasize that performance may vary based on data characteristics, and we cannot assert that using multiple views will always enhance performance. The quality of additional data may also impact the model's overall performance. We have performed an analysis that illustrates the superiority of our proposed model over the SVD approach in all four datasets, with better performance observed in three out of the four datasets.

Furthermore, we conducted an analysis that demonstrates the superiority of our proposed model over the SVD approach in all four datasets, with better performance observed in three out of the four datasets. Additionally, our recommendation model significantly improves the accuracy of recommendations compared to the graph-based recommendation model. It achieves an increase of up to 108% in the Precision metric, a decrease of up to 65.37% in the MAE metric, and a decrease of up to 59.24% in the RMSE metric.

In future work, we will continue to expand by incorporating new sources of information and further improve the Transformer-based model to make the integration of multiple views more efficient.

**Author Contributions:** Conceptualization, T.-L.H.; Data curation, T.-L.H.; Formal analysis, T.-L.H. and A.-C.L.; Investigation, T.-L.H. and A.-C.L.; Methodology, T.-L.H., A.-C.L. and D.-H.V.; Supervision, A.-C.L.; Writing—original draft, T.-L.H.; Writing—review and editing, T.-L.H. and A.-C.L. All authors have read and agreed to the published version of the manuscript.

**Funding:** This research received no external funding.

**Institutional Review Board Statement:** Not applicable.

**Informed Consent Statement:** Not applicable.

**Data Availability Statement:** The datasets analyzed during the current study are available in the web of Grouplens repository (https://grouplens.org/datasets/movielens/ (accessed on 30 October 2022)) and the web of Amazon repository (https://cseweb.ucsd.edu/~jmcauley/datasets/amazon_v2/ (accessed on 30 October 2022)) [54].

**Acknowledgments:** We would like to express our gratitude to Soundararajan Ezekiel from the Department of Mathematics and Computer Science at Indiana University of Pennsylvania, USA, for his valuable discussions, which greatly improved the quality of this paper. He also provided us with significant help in correcting the English writing of the paper.

**Conflicts of Interest:** The authors declare no conflict of interest.

## Abbreviations

The following abbreviations are used in this manuscript:

| | |
|---|---|
| BERT | Bidirectional Encoder Representations from Transformers |
| Bi-LSTM | Bidirectional Long Short-Term Memory |
| CF | Collaborative Filtering |
| DNN | Deep Neural Network |
| GRU | Gated Recurrent Units |
| KNN | K Nearest Neighbors |
| LDA | Latent Dirichlet Allocation |

| MLP | Multi-layer Perceptron |
| RS | Recommender System |
| SVD | Singular Value Decomposition |
| TF-IDF | Term Frequency-Inverse Document Frequency |

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
