# Peer review of "Multiview Fusion Using Transformer Model for Recommender Systems: Integrating the Utility Matrix and Textual Sources"

_applsci, doi:10.3390/app13106324_

Round 1
Reviewer 1 Report
This paper proposes a multi-modal recommender system (RS) based on Transformers. It demonstrates the potential of a multimodal approach for recommendation and highlights the contribution of combining multi-source information to boost recommendations.
The authors correctly stated that there is great interest and value in using multiple sources of information to improve the relevance of recommendations. Interesting recent strategies employ multimodal approaches, which combine data from different sources and harness this joint set of information to improve recommender systems.
Overall, the subject here is very interesting.
The use of English is decent, and the idea is well motivated and presented. The structure of the paper is also good.
Although the list of references seems adequate, it could be further enriched. E.g.,
[1] K. Pliakos and C. Kotropoulos, Simultaneous image tagging and geo-location prediction within hypergraph ranking framework, IEEE International Conference on Acoustics, Speech and Signal Processing (ICASSP), Florence, Italy, 2014, pp. 6894-6898
[2] Xia, X., Yin, H., Yu, J., Wang, Q., Cui, L., & Zhang, X. (2021, May). Self-supervised hypergraph convolutional networks for session-based recommendation. In Proceedings of the AAAI conference on artificial intelligence (Vol. 35, No. 5, pp. 4503-4511).
[3] Fan, W., Ma, Y., Li, Q., He, Y., Zhao, E., Tang, J., & Yin, D. (2019, May). Graph neural networks for social recommendation. In The world wide web conference (pp. 417-426).
The following sentence in the abstract is wrong: “When compared to SVD, which is known as one of the most effective models for recommender systems, the proposed model shows a reduction in MAE from 34.82% to 56.17% for the Amazon datasets”.
There are a few typos left in the manuscript, e.g. line 159 “..”
The authors do not refer to any existing methodologies for multi-modal RSs, such as methods based on hypergraphs or tensors.
RMSE is not reported in Figures 6 and 7.
In Figure 5 the numbers are not displayed.
The authors should clarify the way they tune the hyperparameters of their models (including baselines) as well as disclose all the relevant hyperparameters for each model.
Overall, the greatest weakness in this paper is the evaluation study. Although the authors employed popular benchmark datasets and relevant evaluation metrics, they did not compare their model against any state-of-the-art recommender systems. For example, RSs based on [2], [3], or others included in the related work of this paper.
-
Reviewer 2 Report
The authors propose a Transformer model for recommender systems. Experimental results indicate improvement in recommendations when using Amazon and Movielens databases. Next, this reviewer presents some questions and suggestions to help the authors speed up the work.
Use the term "recommender system" OR "recommendation system" in keywords. You don't need to put both terms.
In the introduction, I suggest the authors add a paragraph describing the research methodology followed in developing the paper.
In related works, the authors explain, at the end of the section, the approach they proposed. However, I expected a more detailed comparison between the works they presented and the research being conducted.
Why did the authors use TF-IDF instead of BERT in the Category-Feature representation?
In Rating-Feature, what are the 'f' characteristics which define the user? Considering that we have an item evaluation matrix, it was unclear to this reviewer what the result of this factoring would be.
After Table 2, the authors need to describe the model training process. For example, how many executions were performed with each model? Are the values presented in Table 4 the average or the best result of each model?
How were the parameters (Layers = 4, Hidden state = 20, Heads = 4) defined?
Some results shown in Table 4 are quite different. However, some are close. Therefore, a statistical analysis is important to be carried out to verify if there is statistical significance in the results.
Minor issues:
- What are the authors considering as a CV? Adding a reference can help clarify the concept, which, in my view, is quite broad. Does this concept apply: "Computer vision is a field of artificial intelligence (AI) that enables computers and systems to derive meaningful information from digital images, videos, and other visual inputs — and take actions or make recommendations based on that information."? [1]
- Equations must be cited in the text.
- I don't know if section 5.2 needs so much detail, as they are metrics that are widely known by the scientific community.
[1] https://www.ibm.com/topics/computer-vision#:~:text=Computer%20vision%20is%20a%20field,recommendations%20based%20on%20that%20information.
Round 2
Reviewer 1 Report
The authors seem to have handled most of my remarks. However, I regret to observe that they have not compared against any state of the art recommender systems. SVD-based RSs, albeit effective, are relatively weak baselines. Although in their cover letter, the authors stated that they compared against GraphRec (an effective rec.sys. that was recently proposed), to my surprise, they just mentioned the performance metrics of the latter model in different datasets. Such values are irrelevant and cannot hold as a comparison, the evaluation should be on the same train-test splits and same datasets.
To this end, I have to insist on my remark about the evaluation process. The authors should compare against a recent recommender system, ideally one that uses multi-modal information. There are plenty of such approaches based on tensors, graphs, hypergraphs, DNNs, or GNNs.
-
